# Examining the Genetic Role of rs8192675 Variant in Saudi Women Diagnosed with Polycystic Ovary Syndrome

**DOI:** 10.3390/diagnostics13203214

**Published:** 2023-10-14

**Authors:** Sarah Alsobaie, Arwa A. Alageel, Tahira Ishfaq, Imran Ali Khan, Khalid Khalaf Alharbi

**Affiliations:** 1Department of Clinical Laboratory Sciences, College of Applied Medical Sciences, King Saud University, Riyadh 11433, Saudi Arabia; salsobaie@ksu.edu.sa (S.A.); aaalageel@ksu.edu.sa (A.A.A.); kharbi@ksu.edu.sa (K.K.A.); 2Department of Obstetrics and Gynecology, College of Medicine, King Saud University, Riyadh 11472, Saudi Arabia; thashmat@ksu.edu.sa

**Keywords:** polycystic ovary syndrome (PCOS), rs8192675 SNP, *SLC2A2* gene, T2DM, Saudi women

## Abstract

Polycystic ovary syndrome is a complex disorder defined by the Rotterdam criteria. Insulin resistance is a common factor for the development of type 2 diabetes mellitus among women with PCOS. The *SLC2A2* gene has been identified as a T2DM gene by genome-wide association studies in the rs8192675 SNP. This study aimed to investigate the rs8192675 SNP in women diagnosed with PCOS on a molecular level and further for T2DM development in the Saudi women. In this case-control study, 100 PCOS women and 100 healthy controls were selected. Among 100 PCOS women, 28 women showed T2DM development. Genotyping for rs8192675 SNP was performed by PCR-RFLP analysis. Additionally, Sanger sequencing was performed to validate the RFLP analysis. The obtained data were used for a statistical analysis for the genotype and allele frequencies, logistic regression, and ANOVA analysis. The clinical data confirmed the positive association between FBG, FI, FSH, TT, TC, HDLc, LDLc, and family histories (*p* < 0.05). HWE analysis was associated in both the PCOS cases and the control individuals. Genotype and allele frequencies were associated in PCOS women and strongly associated with women with PCOS who developed T2DM (*p* < 0.05). No association was found in the logistic regression model or ANOVA analysis studied in women with PCOS (*p* > 0.05). A strong association was observed between the rs8192675 SNP and women with PCOS who developed T2DM using ANOVA analysis (*p* < 0.05). This study confirms that the rs8192675 SNP is associated with women with PCOS and strongly associated with women with PCOS with developed T2DM in Saudi Arabia.

## 1. Introduction

One of the most common endocrine heterogenous disorders among women of reproductive age is polycystic ovary syndrome (PCOS), which involves the reproductive, metabolic, and endocrine systems [1]. Symptoms such as infertility, oligomenorrhea, metabolic disorders, and cutaneous manifestations are connected with PCOS. However, the etiology remains unclear [2]. PCOS is present in 5–10% of women of childbearing age and contributes to 50–70% of anovulatory infertility. Women with PCOS have high androgen secretion, ovulation difficulties, and polycystic ovarian alterations, which can be accompanied by insulin resistance, abdominal obesity, and dyslipidemia [3]. This disorder was first discovered in 1935 by Stein and Leventhal as a group of cases involving women with amenorrhea with polycystic ovaries. The diagnosis of PCOS can be confirmed by (i) NIH, (ii) Rotterdam, and (iii) AE-PCOS society criteria [4]. Nevertheless, the Rotterdam criteria have been recently recommended by health authorities and scientific societies to diagnose patients with PCOS based on hyperandrogenism, polycystic ovaries, and irregular menstrual cycles [5]. Insulin resistance is a common factor for developing type 2 diabetes mellitus (T2DM) in patients with PCOS [6]. Infertility, obesity, irregular menstruation, acne, and hirsutism (a modified Ferriman–Gallway index of eight or higher indicates hirsutism) have been identified as short-term effects in women with PCOS, whereas T2DM, hypertension, certain cancers, coronary heart disease, and sleep are the long-term complications [7]. T2DM is considered as one of the non-modifiable risk factors among women with PCOS [8]. The global prevalence of PCOS has affected up to 20% of reproductive-aged women [9].

Genetic evidence for additional PCOS research was presented by genome-wide association studies (GWAS). Despite the fact that patients with PCOS are accompanied by a number of metabolic concerns, whether or not these conditions contribute to the development of other chronic diseases is unclear [10]. Single-nucleotide polymorphism (SNP) is defined as an amendment in a single-base-pair sequence of DNA linked to the human genome and is connected to GWAS [11]. The biological functions of insulin, including carbohydrate intake and metabolism, glucose synthesis, and lipid metabolism, are impaired in insulin-resistant individuals; elevated insulin levels are required to restore normal metabolism. When insulin resistance is present and pancreatic β-cells are functioning normally, the circulating insulin increases [12]. Solute carrier family 2 member 2 (SLC2A2) is one of the genes connected with the carbohydrate gene, which is associated with GWAS [13]. GLUT2/SLC2A2 variants are predictive of hyperglycemia development, more specifically. A recent study has established a link between T2DM and PCOS with the rs8192675 SNP [14]. The rs8192675 SNP is found in the intron region of the *SLC2A2* gene, which codes for HbA1c effects in T2DM patients. GLUT2 transports glucose to hepatocytes and converts it to glycogen. The GLUT2 protein has 524 amino acid residues. The *SLC2A2* gene is present on chromosome 3q26.2 [15,16].

Only limited studies on women with PCOS in Saudi Arabia have been reported, and currently, no accurate prevalence of women with PCOS exists. However, no robust genetic or molecular studies on Saudi women with PCOS have been documented. Furthermore, the rs8192675 SNP has not been reported. This study will be interesting as the prevalence of chronic diseases in women, including T2DM and obesity, is increasing in Saudi Arabia [17,18,19]. The study aimed to investigate the molecular role of the rs8192675 SNP in the *SLC2A2* gene in women diagnosed with PCOS and observe the development with and without T2DM in Saudi Arabia.

## 2. Methods

### 2.1. Ethical Approval

This study received the ethical grant (E-23-7917) from the Institutional Review Board in the College of Medicine at King Saud University (KSU). All women (*n* = 200) that participated in this study signed the informed consent form, and this study was approved based on the Helsinki Declaration.

### 2.2. Recruitment of PCOS Women

We selected 100 women with PCOS and 100 healthy controls from an outpatient clinic from the Department of Obstetrics and Gynecology at KSU hospital. Both the PCOS and non-PCOS women were selected from the capital city of Saudi Arabia that were within the hospital premises. Samples were recruited after the ethical approval. The inclusion criteria of PCOS women were based on Rotterdam criteria, Saudi-nationality women with an age range of 18–40 years of age. Women without the Rotterdam criteria and with other nationalities were excluded from this study towards the recruitment of PCOS cases. In addition, 100 healthy controls were selected based on normal ovulation and menstruation with the age range of 18–40 years in the Saudi women. Exclusion criteria for healthy controls were women diagnosed with other human diseases, non-Saudi women, and those who did not sign the informed consent form. All women who participated in this study had filled out the questionnaire, and we excluded the premature ovarian insufficiency.

### 2.3. BMI and Blood Analysis

Body mass index (BMI) is described as weight in kilograms (kg) and height in centimeters (cm) or meters squared (m^2^). In this study, we recruited women with normal weight (<24.9 kg/m^2^), overweight women (25.0–29.9 kg/m^2^), with obesity (30.0–34.9 kg/m^2^), morbid obesity-I (35.0–39.9 kg/m^2^), and morbid obesity-II (40.0 kg/m^2^ and above). All 200 women agreed to donate 5 mL of peripheral blood towards this research. The blood was bifurcated into 3 mL for biochemical analysis and 2 mL for molecular analysis.

### 2.4. Biochemical Analysis

Fasting blood glucose (FBG), fasting insulin (FI), serum creatinine, follicle-stimulating hormone (FSH), luteinizing hormone (LH), thyroid-stimulating hormone (TSH), total testosterone (TT), aspartate aminotransferase (AST), alanine transaminase (ALT), and lipid profile parameters, including total cholesterol (TC), triglycerides (TG), and high- and low-density lipoprotein cholesterol (HDLc/LDLc) levels were analyzed in serum from women with PCOS and control women.

### 2.5. Molecular Analysis for rs8192675 SNP

Genomic DNA was extracted using a Qiagen DNA isolation kit according to the protocol. Extracted genomic DNA was quantified using NanoDrop spectrophotometer to measure the DNA quality. All the DNA quality was converted into 20 µg/mL and stored at −80 °C. Genotyping for rs8192675 SNP was performed using polymerase chain reaction (PCR) with the following primers: forward: GGGTTCATCCTTCCAGTGAA and reverse: AAACCCAGGCAGTCAAACAC. Moreover, 50 µL of reaction mixture using Qiagen Master Mix (20 µL), 10 pmol of primer set (4 µL), 22 µL of double distilled water, and 4 µL of genomic DNA were used. PCR was performed with initial denaturation (at 95 °C for 5 min), denaturation (at 95 °C for 30 s), annealing (at 66 °C for 30 s), extension (at 72 °C for 45 s), and final extension (at 72 °C for 5 min); after 35 cycles, samples were held at 4 °C. The undigested PCR product of 619 bp (AA genotype) was digested towards restriction fragment length polymorphism (RFLP) analysis using ACC651 restriction enzyme at the site (A/G^↑^GTACC) to convert into 323/296 (GG genotype). The heterozygous genotype was found to be the combination of AA and GG genotypes, i.e., 619/323/296 bp (Figure 1). The digestion was performed for 18 h at 37 °C using 5U of restriction enzyme. Both digested and undigested PCR products were run on 2% and 3% agarose gels stained with ethidium bromide. Additionally, 9% of PCR products were validated for Sanger sequencing (Figure 2) to reconfirm the RFLP analysis.

### 2.6. Statistical Analysis

Categorical variables were documented as total numbers and percentages; numerical variables were documented as means and standard deviations. Independent sample *t*-test was calculated for statistical association between PCOS cases and controls (Table 1). Hardy Weinberg Equilibrium (HWE; Table 2) was tested between PCOS cases and controls using an Excel sheet. Genotype and allele frequencies were measured between PCOS cases and controls (Table 3) and T2DM women in PCOS cases and controls (Table 4) using SNPstats software (https://www.snpstats.net/start.htm (accessed on 10 September 2023)) by odds ratios and 95% confidence intervals (CI). Logistic regression analysis (Table 5) was measured between dependent variables and rs8192675 SNP using SPSS software version 27.0. One-way ANOVA analysis (Table 6 and Table 7) was measured using Jamovi software (https://www.jamovi.org/ (accessed on 10 September 2023)) for rs8192675 SNPs and PCOS and for T2DM in patients with PCOS. Statistical association was confirmed using *p* < 0.05 between 2–3 groups. The prevalence of family histories in both the control and women with PCOS was represented by Origin software (Version 9.9) (Figure 3).

## 3. Results

### 3.1. Characteristics between Women with PCOS and Control Individuals

The clinical and biological characteristics of the women with PCOS and the control individuals are highlighted in Table 1. Among the 100 women with PCOS, the mean age was found to be lower (30.28 ± 5.83) when compared with that of the control individuals (31.39 ± 6.71). Weight levels were elevated and BMI was found to be high in the control individuals in comparison to those in patients with PCOS. FBG, FI, FSH, TT, TC, HDLc, and LDLc were strongly correlated when comparing PCOS cases and controls (*p* < 0.05). However, age, height, LH, TSH, and TG were not correlated (*p* > 0.05). Family history of PCOS was strongly associated when comparing the PCOS and control subjects (*p* < 0.0001).

### 3.2. HWE Analysis in rs8192675 SNP

The distribution of genotype frequencies for rs8192675 SNP was found to be consistent towards HWE analysis for both the control (*p* = 0.66) and PCOS cases (*p* = 0.55). The samples obtained in this study were in genetic equilibrium (Table 2).

### 3.3. Genotype and Allele Frequency Studies in Women with PCOS with rs8192675 SNP

In this case-control study, genotype and allele frequencies were studied among 100 women with PCOS and 100 control women (Table 3). The genotype frequencies in women with PCOS for the AA, AG, and GG genotypes were found to be 52%, 39%, and 9%, whereas in control women they were 69%, 27%, and 4%, respectively. The A and G allele frequencies were 71.5% and 28.5% in women with PCOS and 82.5% and 17.5% in the controls. The genotypes (AG vs. AA; OR-1.19; 95% CI (1.04–3.52); *p* = 0.03), genetic models (AG + GG vs. AA; OR-2.05; 95% CI (1.15–3.66); *p* = 0.01), and allele frequencies (G vs. A; OR-1.87; 95% CI (1.16–3.02); *p* = 0.008) showed the statistical association when compared between the women with PCOS and the controls. However, other genetic models showed no association between the groups (GG vs. AA; OR- 2.98; 95% CI (0.87–10.23); *p* = 0.07, AA + GG vs. AG; OR-0.57; 95% CI (0.31–1.050; *p* = 0.07 and AA + AG vs. GG; OR-0.42; 95% CI (0.12–1.41); *p* = 0.15).

### 3.4. Genotype and Allele Frequency Studies in Patients with T2DM among Women with PCOS with rs8192675 SNP

Both genotype and allele frequencies for T2DM patients in women with PCOS and the controls for rs8192675 SNP are shown in Table 4. The genotype frequency for the rs8192675 SNP of the AA, AG, and GG genotypes in T2DM patients in PCOS women were 28.6%, 50%, and 21.4%; and among the controls, they were found to be 69%, 27%, and 4%, respectively. The A and G allele frequencies were 53.6% and 46.4% in patients with T2DM among women with PCOS and 82.5% and 17.5% in the controls. The G allele and the GG genotype frequency were significantly different between both groups, indicating that the G allele and GG genotypes conferred a higher genetic risk (OR-4.08 (95% CI: 2.15–7.74); *p* < 0.0001 and OR-12.98 (95% CI: 3.0–5.58); *p* = 0.0001). Additionally, the dominant model (AG + GG vs. AA: OR-5.56 (95% CI: 2.21–14.0); *p* = 0.0001) and the heterozygous genotype (AG vs. AA: OR-4.47 (95% CI: 1.68–11.87); *p* = 0.001) were also strongly associated between both groups.

### 3.5. Logistic Regression Model with Dependent and Independent Variables in rs8192675 SNP

The dependent variables involved in Table 5 are documented as clinical characteristics, such as age, weight, BMI, FBG, FI, FSH, LH, TSH, TT, TC, TG, HDL-c, and LDLc as dependent variables and the rs8192675 SNP as an independent variable. The single logistic regression model could not confirm any positive association with any of the single dependent variables using the independent variable. The analysis of this study confirmed the negative association when comparing the dependent and independent variables in this study.

### 3.6. ANOVA Analysis in Women with PCOS

ANOVA analysis was performed between the rs8192675 genotypes and the women with PCOS, as shown in Table 6. The three genotypes were categorized as AA (*n* = 52), AG (*n* = 39), and GG (*n* = 9) based on the obtained values performed based on PCR-RFLP analysis. The dependent variables were age, weight, BMI, FBG, FI, FSH, LH, TSH, TT, and the lipid profile parameters. The elevated levels found in the AA genotypes were the LH (7.54 ± 4.25), TC (5.17 ± 1.32), and HDLc (0.68 ± 0.32) variables. Weight (73.87 ± 12.30), FBG (5.30 ± 1.13), FI (12.78 ± 7.79), and LDLc (3.78 ± 0.86) levels had elevated levels in the AG genotypes, and in the GG genotypes, age (31.56 ± 5.36), BMI (29.44 ± 3.70), FSH (7.71 ± 2.97), TSH (2.37 ± 0.65), TT (2.04 ± 0.89), and TG (2.14 ± 1.37) levels were elevated. The current results show no association of genotypes with any of the dependent variables in PCOS women.

### 3.7. ANOVA Analysis in T2DM Patients in PCOS Women

Table 7 defines the ANOVA analysis studied between the rs8192765 genotypes and in 28 women with PCOS who developed T2DM. The elevated levels were identified in the AA and AG genotypes for each of the five and in the GG genotypes for three parameters. The elevated levels found in the AA genotypes were in weight (79.59 ± 14.55), BMI (31.91 ± 6.35), LH (9.17 ± 6.50), TC (5.30 ± 1.68), and TG (2.71 ± 1.99); whereas in the AG genotypes, abnormal levels were found in age (32.14 ± 5.42), FBG (5.40 ± 1.50), FI (13.29 ± 7.87), HDLc (0.70 ± 0.20), and LDLc (3.59 ± 1.04) levels. However, abnormal FSH (8.75 ± 3.17), TSH (2.61 ± 0.64), and TT (2.01 ± 0.85) levels were found to be in GG genotypes. ANOVA analysis confirmed statistical associations in weight (*p* = 0.01), BMI (*p* = 0.0008), FBG (*p* = 0.03), FSH (*p* = 0.0001), TT (*p* = 0.04), and TG (*p* = 0.001).

## 4. Discussion

There is a huge demand towards rapid and accurate genotyping for various ethnic backgrounds to address chronic etiologies, such as PCOS, diabetes, obesity, and other human diseases. PCR technique was found to be convenient towards genotyping [20]. In this study, we tried to evaluate the possible association between women with PCOS and the rs8192675 SNP in the *SLC2A2* gene in Saudi Arabia. The rs8192675 SNP is associated with one of the carbohydrate genes, which was identified via GWAS. The facilitated glucose transporter GLUT2 is encoded by the C allele of rs8192675 in the intron region [13]. Furthermore, the current era has seen an increase in chronic disorders, particularly in T2DM and obesity in Saudi Arabia [21]. We observed a positive role of genotype and allele frequencies with the rs8192675 SNP in the *SLC2A2* gene and a strong association between women with PCOS who developed T2DM (*n* = 28) when compared with the controls (*p* < 0.05). A logistic regression model and ANOVA analysis in women with PCOS showed a negative association (*p* > 0.05). However, the results of the ANOVA analysis showed a positive association when compared with women with PCOS who developed T2DM in the rs8192675 SNP (*p* < 0.05).

The role of the rs8192675 SNP had not been studied in any of the diseases apart from diabetes as this SNP and gene is connected with the regulatory effect of metformin [22]. In Saudi Arabia, the rs8192675 SNP is already documented with T2DM [15]. Hence, we decided to study women with PCOS as there is a connection between T2DM and PCOS [23,24,25]. Insulin resistance is the intermediate element in women with PCOS for the development of T2DM. Though PCOS women produce enough insulin, they are unable to use it properly, resulting in a rise in T2DM [26]. In this study, we could not document the accurate usage of medication in PCOS women, which could be one of the limitations of this study. When compared to women without PCOS, women with PCOS have a 2–3-fold increased prevalence of prediabetes and T2DM and a fourfold increased risk of developing obstructive sleep apnea [27].

In T2DM patients, SNP rs8192675 in the *SLC2A2* gene was related with a better glucose response. The rs8192675 SNP was documented in the global studies, and both positive and negative associations were documented within different ethnicities [13,15,16,28,29,30,31,32]. None of the meta-analysis studies have been focused towards rs8192675 and T2DM. However, the rs8192675 SNP has not been studied in any disease apart from T2DM [33,34,35]. The rs8192675 SNP was studied in the Saudi population from the northwestern region of Saudi Arabia, and in this study, the authors selected 100 T2DM patients and 100 healthy controls within the Tabuk region. The study results confirmed that the AA (TT), AG (TC), and GG (CC) genotypes represented 22%, 75%, and 3% of the T2DM patients. The control subjects documented 36%, 58%, and 6% in the rs8192675 SNP. However, in our study, the varied genotypes were documented (Table 3 and Table 4) as we opted for women subjects only within the age range of 18–40 years of age. However, the authors from the Almutairi et al. studies did not document the ages among their studies [15]. Finally, our study was conducted in the central region and capital city of Saudi Arabia.

In this study, the control women were found to have high levels of weight (77.56 ± 11.86) and BMI (30.68 ± 4.53) when compared with those of the PCOS women (73.78 ± 11.50 and 29.26 ± 4.87). In fact, the women with PCOS were considered to be overweight, and normal women were found to be obese. This is due to the fact that obesity prevalence has increased over a decade, and women were found to be obese when compared with male individuals in the kingdom. The prevalence of normal BMI levels in women with PCOS was found to be 19% while 12% was documented in the controls; overweight prevalence was found to be 38% in PCOS women and 30% in the controls; whereas obesity was highly documented at 39% in the controls and 33% in the PCOS women. Finally, morbid obesity-I was found to be high in the controls (17%) when compared with that in the PCOS women (8%). Morbid obesity-II was found to be similar in both of the groups with 2% each. Additionally, age was found to be higher in the controls (31.39 ± 6.71) than among the PCOS women (30.28 ± 5.83). This is mostly due to two factors: the selection of exclusively Saudi women and the signing of the consent form to participate in this study. The control women did not have a family history of infertility or PCOS.

We tried to explore the previous studies in which PCOS women developed diabetes, specifically T2DM. Forsulund et al. [36] confirmed that 19% of the PCOS women and 1% of the non-PCOS women developed T2DM, and additionally, elevated BMI had also played a role. One of the previous studies from Saudi Arabia confirmed that 15.7% of adolescent women diagnosed with T1DM developed PCOS [37]. A review article by Aljulifi [17] confirmed that PCOS women had a higher risk of developing diabetes [5]. One of the previous studies in Saudi Arabia documented that the mean HbA1c levels in PCOS women was found to be 6.15 ± 2.31, which is an indication of prediabetes among the 31 patients in their study [38]. The majority of documented studied in PCOS women have not described any other disease apart from obesity. The overall conclusion indicates that the majority of the studies were not focused on any other human diseases in PCOS women.

One of the limitations of this study could be not documenting the medication usage in all the PCOS women. Another limitation of this study could be the screening of a single variant, and a final limitation of this study could be not following up with the PCOS women. The strength of this study was enrolling all Saudi women and performing the validation for this study via Sanger sequencing.

## 5. Conclusions

This study concludes that the rs8192675 SNP was associated with women suffering from PCOS; a stronger association was found in PCOS-suffering women who developed diabetes. However, the study is based on only 28 diabetic women. Further screenings of additional SNPs in *SLC2A2* are needed to observe the biomarker for T2DM disease in the Saudi population. Additionally, meta-analysis studies should also be performed between the rs8192675 SNP and T2DM in the global ethnicity.

## Figures and Tables

**Figure 1 diagnostics-13-03214-f001:**
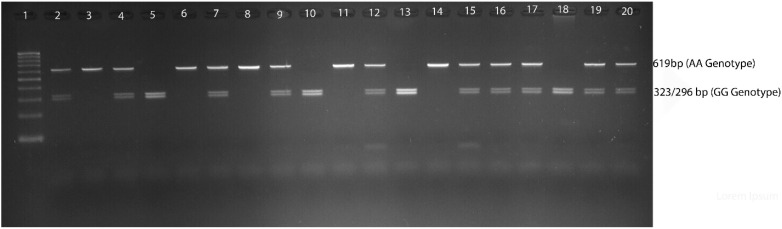
The 3% agarose gel represents rs8192675 variants in PCOS women.

**Figure 2 diagnostics-13-03214-f002:**
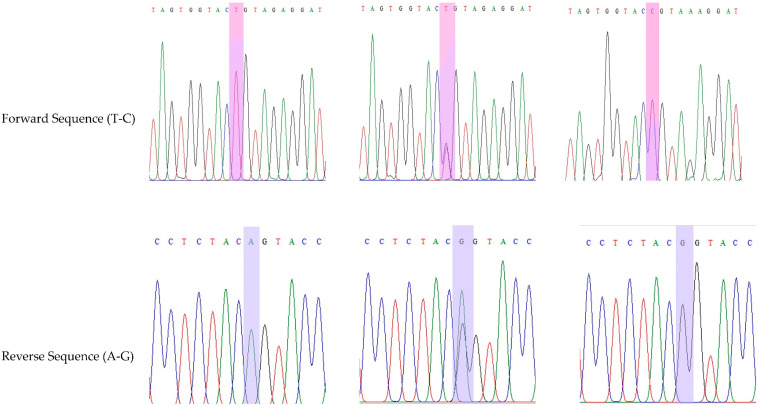
Validation was performed in Sanger sequencing analysis in rs8192675 variants.

**Figure 3 diagnostics-13-03214-f003:**
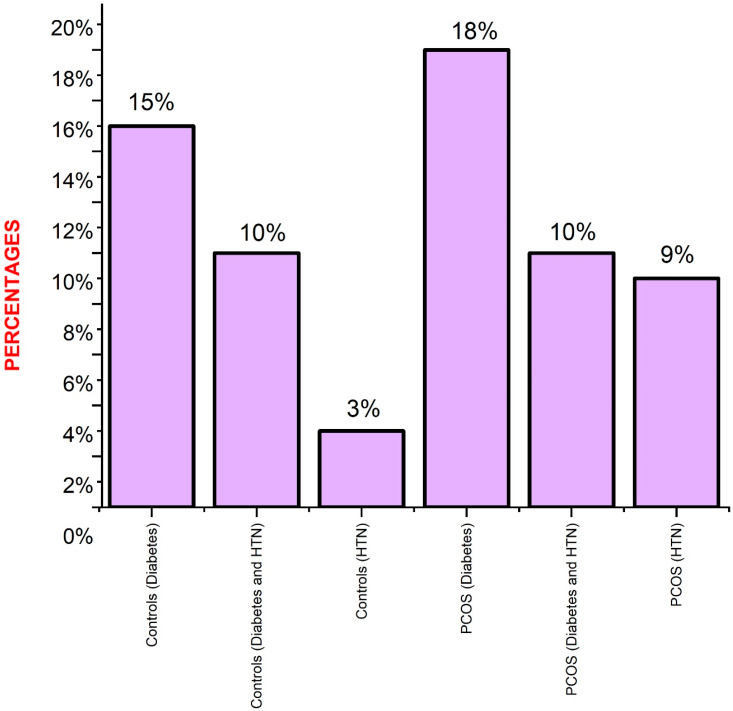
Representation of family history of chronic diseases in PCOS women and controls.

**Table 1 diagnostics-13-03214-t001:** Clinical details studied between PCOS and control women.

Women Characteristics	PCOS (*n* = 100)	Controls (*n* = 100)	*p* Value
Age (years)	30.28 ± 5.83	31.39 ± 6.71	0.21
Weight (kgs)	73.78 ± 11.50	77.56 ± 11.86	0.02
Height (cms)	158.81 ± 5.16	158.21 ± 6.88	0.48
BMI (kg/m^2^)	29.26 ± 4.87	30.68 ± 4.53	0.03
FBG (mmol/L)	5.06 ± 0.87	4.72 ± 0.72	0.002
Fasting insulin (µIU/mL)	11.03 ± 6.37	8.95 ± 5.14	0.01
FSH (IU/mL)	6.85 ± 2.71	6.08 ± 2.42	0.03
LH (IU/mL)	7.31 ± 4.24	6.95 ± 2.35	0.45
TSH (mIU/L)	2.23 ± 0.86	2.12 ± 0.76	0.33
Total testosterone (nmol/L)	1.91 ± 0.87	0.98 ± 0.81	<0.0001
TC (mmol/L)	5.14 ± 1.13	3.09 ± 0.38	<0.0001
TG (mmol/L)	1.84 ± 1.15	1.59 ± 0.88	0.08
HDLc (mmol/L)	0.67 ± 0.29	0.48 ± 0.14	<0.0001
LDLc (mmol/L)	3.70 ± 0.98	3.19 ± 0.56	0.0001
Other family histories	37 (37%)	28 (28%)	<0.0001
Family history of PCOS	28 (28%)	0 (0)	<0.0001

**Table 2 diagnostics-13-03214-t002:** HWE analysis studied with rs8192675 SNP in PCOS and control women.

	PCOS Women	Control Subjects
HWE analysis	0.29	0.18
χ^2^	0.18	0.42
*p* value	0.66	0.51

**Table 3 diagnostics-13-03214-t003:** Genotype and allele frequency studies in rs8192675 SNP between PCOS and control women.

Genotype and Allele	Controls (*n* = 100)	PCOS (*n* = 100)	OR (95%CI)	*p* Value
AA Genotype	69 (69.0%)	52 (52.0%)	Reference	Reference
AG Genotype	27 (27.0%)	39 (39.0%)	1.19 (1.04–3.52)	0.03
GG Genotype	04 (4.0%)	09 (9.0%)	2.98 (0.87–10.23)	0.07
AG + GG vs. AA	31 (31.0%)	48 (48.0%)	2.05 (1.15–3.66)	0.01
AA + GG vs. AG	73 (73.0%)	61 (61.0%)	0.57 (0.31–1.05)	0.07
AA + AG vs. GG	96 (96.0%)	91 (91.0%)	0.42 (0.12–1.41)	0.15
A Allele	165 (82.5%)	143 (71.5%)	Reference	Reference
G Allele	35 (17.5%)	57 (28.5%)	1.87 (1.16–3.02)	0.008

**Table 4 diagnostics-13-03214-t004:** Genotype and allele frequency studies in rs8192675 SNP between patients with T2DM among women with PCOS and controls.

Genotype and Allele	T2DM + PCOS (*n* = 28)	Controls (*n* = 100)	OR (95% CI)	*p* Value
AA Genotype	08 (28.6%)	69 (69.0%)	Reference	Reference
AG Genotype	14 (50.0%)	27 (27.0%)	4.47 (1.68–11.87)	0.001
GG Genotype	06 (21.4%)	04 (4.0%)	12.98 (3.0–55.8)	0.0001
AG + GG vs. AA	20 (71.4%)	31 (31.0%)	5.56 (2.21–14.0)	0.0001
AA + GG vs. AG	14 (50.0%)	73 (73.0%)	0.36 (0.15–0.87)	0.02
AA + AG vs. GG	22 (78.6%)	96 (96.0%)	0.15 (0.03–0.58)	0.002
A Allele	30 (53.6%)	165 (82.5%)	Reference	Reference
G Allele	26 (46.4%)	35 (17.5%)	4.08 (2.15–7.74)	<0.0001

**Table 5 diagnostics-13-03214-t005:** Multiple logistic regression analysis performed between rs8192675 SNP and women with PCOS.

Dependent Variables	R-Value	Adjusted R Square	F	*p* Value
Age (years)	0.029	−0.009	0.083	0.773
Weight (kgs)	0.014	−0.10	0.018	0.893
BMI (kg/m^2^)	0.032	−0.009	0.100	0.752
FBG (mmol/L)	0.114	0.003	1.297	0.258
Fasting insulin (µIU/mL)	0.054	−0.007	0.283	0.596
FSH (IU/mL)	0.027	−0.009	0.070	0.792
LH (IU/mL)	0.110	0.002	1.211	0.274
TSH (mIU/L)	0.071	−0.005	0.498	0.482
Total testosterone (nmol/L)	0.024	−0.010	0.054	0.816
TC (mmol/L)	0.046	−0.008	0.206	0.651
TG (mmol/L)	0.044	−0.008	0.189	0.664
HDLc (mmol/L)	0.101	0.000	1.001	0.320
LDLc (mmol/L)	0.039	−0.009	0.146	0.703

**Table 6 diagnostics-13-03214-t006:** ANOVA analysis in rs8192675 genotypes with PCOS dependent variables.

Dependent Variables	AA (*n* = 52)	AG (*n* = 39)	GG (*n* = 9)	*p* Value
Age (years)	30.29 ± 6.22	29.97 ± 5.50	31.56 ± 5.36	0.76
Weight (kgs)	73.85 ± 11.25	73.87 ± 12.30	73.04 ± 10.42	0.97
BMI (kg/m^2^)	29.09 ± 4.97	29.42 ± 5.06	29.44 ± 3.70	0.94
FBG (mmol/L)	4.90 ± 0.61	5.30 ± 1.13	4.88 ± 0.67	0.07
Fasting insulin (µIU/mL)	10.14 ± 5.34	12.78 ± 7.79	8.46 ± 1.61	0.06
FSH (IU/mL)	6.91 ± 2.93	6.58 ± 2.34	7.71 ± 2.97	0.99
LH (IU/mL)	7.54 ± 4.25	7.45 ± 4.46	5.31 ± 2.87	0.33
TSH (mIU/L)	2.18 ± 0.80	2.27 ± 0.99	2.37 ± 0.65	0.78
Total testosterone (nmol/L)	1.91 ± 0.83	1.88 ± 0.95	2.04 ± 0.89	0.88
TC (mmol/L)	5.17 ± 1.32	5.15 ± 0.92	4.92 ± 0.67	0.82
TG (mmol/L)	1.96 ± 1.31	1.62 ± 0.80	2.14 ± 1.37	0.27
HDLc (mmol/L)	0.68 ± 0.32	0.67 ± 0.26	0.56 ± 0.26	0.52
LDLc (mmol/L)	3.70 ± 1.09	3.78 ± 0.86	3.39 ± 0.76	0.56

**Table 7 diagnostics-13-03214-t007:** ANOVA analysis in rs8192675 genotypes and PCOS women with T2DM.

Dependent Variables	AA (*n* = 08)	AG (*n* = 14)	GG (*n* = 16)	*p* Value
Age (years)	31.38 ± 7.09	32.14 ± 5.42	31.33 ± 4.08	0.83
Weight (kgs)	79.59 ± 14.55	71.14 ± 13.93	71.60 ± 8.98	0.01
BMI (kg/m^2^)	31.91 ± 6.35	27.36 ± 4.72	29.40 ± 2.81	0.0008
FBG (mmol/L)	4.86 ± 0.77	5.40 ± 1.50	4.58 ± 0.56	0.03
Fasting insulin (µIU/mL)	11.16 ± 5.97	13.29 ± 7.87	8.83 ± 1.84	0.12
FSH (IU/mL)	6.45 ± 1.18	5.67 ± 1.29	8.75 ± 3.17	0.0001
LH (IU/mL)	9.17 ± 6.50	7.68 ± 4.84	5.78 ± 3.48	0.18
TSH (mIU/L)	2.37 ± 0.57	2.09 ± 0.96	2.61 ± 0.64	0.08
Total testosterone (nmol/L)	1.60 ± 0.39	1.89 ± 0.80	2.01 ± 0.85	0.04
TC (mmol/L)	5.30 ± 1.68	4.97 ± 1.03	4.88 ± 0.83	0.46
TG (mmol/L)	2.71 ± 1.99	1.56 ± 0.77	1.59 ± 1.00	0.001
HDLc (mmol/L)	0.65 ± 0.32	0.70 ± 0.20	0.60 ± 0.30	0.53
LDLc (mmol/L)	3.58 ± 1.02	3.59 ± 1.04	3.56 ± 0.82	0.99

## Data Availability

Not applicable.

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
