# Peer review of "Examining the Genetic Role of rs8192675 Variant in Saudi Women Diagnosed with Polycystic Ovary Syndrome"

_diagnostics, 2023, doi:10.3390/diagnostics13203214_

Round 1

Reviewer 1 Report

The association between PCOS and diabetes is a clinically important topic. This study increases our knowledge about the mechanisms behind the association. I believe the sample size is sufficient. The methods used are fine.

Abstract

Clinical data confirmed the positive association between FBG, FI, FSH, TT, TC, HDLc, LDLc and family histories.

Should be associations and please do not use abbreviations in the abstract as several of the readers will not know what the abbreviations stand for.

Introduction

PCOS diagnosis can be confirmed by (i) NIH, (ii) Rotterdam and (iii) AE-PCOS society [4].

The diagnosis is confirmed by criteria defined according to these three categories. Please rephrase.

Methods

In the Discussion the authors state that the patients and controls are from within Tabuk region. Please insert also in the Methods section as it is good to know that the patients and controls are from the same region. I assume that the same exclusion criteria (other nationalities) also apply to the control group?

Table 2

Could you change the text level for the two columns so that they are even vertically.

Table 6 and 7

Please rephrase the headings: Anova analysis studied and Anova analysis was studied is not optimal. Please rephrase

Results

Elevated weight levels (77.56 ± 11.86 vs 73.78 ± 11.50; p=0.02) and BMI

Please omit levels

Table 5 shows a logistic regression model studied with clinical characteristics,

Please rephrase

Discussion

rs8192675 SNP had not been studied in any of the disease apart from diabetes

Please rephrase: rs8192675 SNP had not been studied in any disease apart from diabetes

Author Response

The association between PCOS and diabetes is a clinically important topic. This study increases our knowledge about the mechanisms behind the association. I believe the sample size is sufficient. The methods used are fine.

A) Thank You for your valuable comment.

Abstract

Clinical data confirmed the positive association between FBG, FI, FSH, TT, TC, HDLc, LDLc and family histories. Should be associations and please do not use abbreviations in the abstract as several of the readers will not know what the abbreviations stand for.

A) We have updated the abstract in the revised manuscript.

Introduction

PCOS diagnosis can be confirmed by (i) NIH, (ii) Rotterdam and (iii) AE-PCOS society [4].

The diagnosis is confirmed by criteria defined according to these three categories. Please rephrase.

A) We have rephrased your suggestion and added in the revised manuscript.

Methods

In the Discussion the authors state that the patients and controls are from within Tabuk region. Please insert also in the Methods section as it is good to know that the patients and controls are from the same region. I assume that the same exclusion criteria (other nationalities) also apply to the control group?

A) We have added in the methodology section. Thank You Reviewer for your valuable comment. We have described in detail about the inclusion and exclusion criteria of PCOS and control women.

Table 2

Could you change the text level for the two columns so that they are even vertically.

A) We have updated in the revised manuscript.

Table 6 and 7

Please rephrase the headings: Anova analysis studied and Anova analysis was studied is not optimal. Please rephrase

A) We have rephrased the recommended sentence in the revised manuscript.

Results

Elevated weight levels (77.56 ± 11.86 vs 73.78 ± 11.50; p=0.02) and BMI

Please omit levels

A) We have removed the values of weight and BMI in the revised manuscript.

Table 5 shows a logistic regression model studied with clinical characteristics,

Please rephrase

A) We have edited the title in the table 5. Thank you for this comment.

Discussion

rs8192675 SNP had not been studied in any of the disease apart from diabetes

Please rephrase: rs8192675 SNP had not been studied in any disease apart from diabetes

A) We have rephrased as per your recommendation.

